# Usnic Acid and *Usnea barbata* (L.) F.H. Wigg. Dry Extracts Promote Apoptosis and DNA Damage in Human Blood Cells through Enhancing ROS Levels

**DOI:** 10.3390/antiox10081171

**Published:** 2021-07-23

**Authors:** Violeta Popovici, Elena Matei, Georgeta Camelia Cozaru, Mariana Aschie, Laura Bucur, Dan Rambu, Teodor Costache, Iulia Elena Cucolea, Gabriela Vochita, Daniela Gherghel, Aureliana Caraiane, Victoria Badea

**Affiliations:** 1Department of Microbiology and Immunology, Faculty of Dental Medicine, Ovidius University of Constanta, 7 Ilarie Voronca Street, 900684 Constanta, Romania; violeta.popovici@365.univ-ovidius.ro (V.P.); victoria.badea@365.univ-ovidius.ro (V.B.); 2Center for Research and Development of the Morphological and Genetic Studies of Malignant Pathology, Ovidius University of Constanta, CEDMOG, 145 Tomis Blvd., 900591 Constanta, Romania; drcozaru@yahoo.com (G.C.C.); aschiemariana@yahoo.com (M.A.); 3Clinical Service of Pathology, Sf. Apostol Andrei Emergency County Hospital, 145 Tomis Blvd., 900591 Constanta, Romania; 4Department of Pharmacognosy, Faculty of Pharmacy, Ovidius University of Constanta, 6 Capitan Al. Serbanescu Street, 900001 Constanta, Romania; 5Research Center for Instrumental Analysis SCIENT, 1E Petre Ispirescu Street, Tancabesti, 077167 Ilfov, Romania; dan.rambu@scient.ro (D.R.); teodor.costache@scient.ro (T.C.); iulia.cucolea@scient.ro (I.E.C.); 6NIRDBS, Institute of Biological Research Iasi, 47 Lascar Catargi Street, 700107 Iasi, Romania; gabriela.vochita@icbiasi.ro (G.V.); daniela.gherghel@icbiasi.ro (D.G.); 7Department of Oral Rehabilitation, Faculty of Dental Medicine, Ovidius University of Constanta, 7 Ilarie Voronca Street, 900684 Constanta, Romania; aureliana.caraiane@365.univ-ovidius.ro

**Keywords:** *Usnea barbata*, usnic acid, secondary metabolites, blood cells, DNA damage, apoptosis, cytotoxic effect, oxidative stress

## Abstract

Nowadays, numerous biomedical studies performed on natural compounds and plant extracts aim to obtain highly selective pharmacological activities without unwanted toxic effects. In the big world of medicinal plants, *Usnea barbata* (L) F.H. Wigg (*U. barbata*) and usnic acid (UA) are well-known for their therapeutical properties. One of the most studied properties is their cytotoxicity on various tumor cells. This work aims to evaluate their cytotoxic potential on normal blood cells. Three dry *U. barbata* extracts in various solvents: ethyl acetate (UBEA), acetone (UBA), and ethanol (UBE) were prepared. From UBEA we isolated usnic acid with high purity by semipreparative chromatography. Then, UA, UBA, and UBE dissolved in 1% dimethyl sulfoxide (DMSO) and diluted in four concentrations were tested for their toxicity on human blood cells. The blood samples were collected from a healthy non-smoker donor; the obtained blood cell cultures were treated with the tested samples. After 24 h, the cytotoxic effect was analyzed through the mechanisms that can cause cell death: early and late apoptosis, caspase 3/7 activity, nuclear apoptosis, autophagy, reactive oxygen species (ROS) level and DNA damage. Generally, the cytotoxic effect was directly proportional to the increase of concentrations, usnic acid inducing the most significant response. At high concentrations, usnic acid and *U. barbata* extracts induced apoptosis and DNA damage in human blood cells, increasing ROS levels. Our study reveals the importance of prior natural products toxicity evaluation on normal cells to anticipate their limits and benefits as potential anticancer drugs.

## 1. Introduction

Natural products have a significant role in modern drug development, especially as antitumor agents [1]. Since discovering that plant secondary metabolites have been elaborated for adaptive reasons [2] within living systems [3], they are often understood as exhibiting more drug-likeness and biological friendliness [4] than totally synthetic molecules [5]. Complex biomedical studies performed on isolated natural compounds and plant extracts aim to obtain high therapeutic activity to treat various diseases without unwanted effects [6]. Especially in oncological pathology, in vitro and in vivo studies have as their principal objective a selective cytotoxic action against tumor cells without affecting the normal ones [7].

In the big world of medicinal plants, lichens are symbiotic organisms [8] between a fungus and microalgae/cyanobacteria [9], known since ancient times for their biological effects [10]. As an important representative of this plant group, *Usnea barbata* (L) F.H. Wigg (*U. barbata*) is a fruticose thalli lichen with interesting therapeutic properties [11]; this species has been used for thousands of years in traditional medicine worldwide to treat various diseases [12]. The wide range of bio-activities (antioxidant [13], antimicrobial [14], anti-inflammatory [15], anticancer [16], cytotoxic [17], pro-oxidant [18]) is due to the content of active secondary metabolites [19] synthesized by the mycobiont (lichen-forming fungus) [20]. The phytochemical profile of *U. barbata* is already known. The metabolomics of this species belongs to different classes of chemical compounds: depsides (barbatic acid, methyl-8-hydroxy-4-*O*-demethylbarbatate, baeomycesic acid, 8-hydroxybarbatic acid), depsidones (connorstictic acid, fumarprotocetraric acid, hypoconstictic acid, lobaric acid), lipids (polyhydroxylated lipids), and dibenzofurans (usnic acid, placodiolic acid) [21]. Of all these lichen secondary metabolites, usnic acid [22] is by far the best known [23] and responsible for most bio-activities [24] of the *U. barbata* and, at the same time, of all lichens of the *Usnea* genus [25].

Usnic acid is an extensively studied lichen metabolite with controversial [26,27] results related to its benefits in relationship with the extraction method and the lichen species [28]. It was used to induce human weight loss [29], although unwanted hepatotoxic effects were also triggered [30]. In addition, UA highlights antimicrobial [31], insecticidal [32], anticholinergic [33], antioxidant [34], pro-oxidant [35], antigenotoxic [36], genotoxic [37], teratogenic [38], anti-inflammatory [39], analgesic and antipyretic [40], mutagenic and carcinogenic [41], anticancer [42], and cytotoxic [43] activities. Numerous researchers have shown the pharmacological actions of usnic acid and *Usnea* sp. extracts, especially cytotoxic activity, on different types of tumor cells [44].

The most important event resulting from the cytotoxic activity is cell death, which consists of morphological alterations [45]. Hence, the highly described mechanism in usnic acid and *Usnea* sp. anticancer activity is apoptosis [46]. This programmed cell death (PCD) is associated with DNA fragmentation and recognized by morphological characteristics as well as cytoplasmic condensation, nuclear pyknosis, chromatin condensation, cell rounding, membrane blebbing, and cytoskeletal collapse. In addition, membrane-bound apoptotic bodies are formed; macrophages rapidly digest them without activating the immune response [47]. In apoptosis, biochemical events through two distinct pathways (extrinsic and intrinsic) are correlated with these morphological changes [48]. Thus, the common extrinsic pathway (receptor-mediated) begins with receptor binding and activation of the initiator caspase-8. The following step is caspase-3 activation by caspase-8 or Bid-a B-cell lymphoma 2 (Bcl-2) pro-apoptotic protein-cleavage. Bid splitting brings mitochondrial cytochrome *c* leakage and apoptosomes formation. The intrinsic (mitochondrial) apoptotic pathway consists of cytochrome *c* release by Bcl-2 pro-apoptotic proteins action. Next, cytochrome *c* interacts with Apaf-1, dATP, and procaspase 9, generating apoptosomes. As a result, caspase-9 and -3 activation follows in both pathways. Moreover, various cell apoptosis can occur through common or specific biochemical processes [48,49]; for instance, a considerable diversity of molecular mechanisms involved in this PCD was highlighted in the different blood cells types. Thereby, nucleus-free platelets exhibit increasing mitochondrial functions in ATP synthesis, energy metabolism, cells survival and apoptosis activation [49]. Thereby, the major apoptotic pathway is the intrinsic one, with overexpression of Bcl-2 pro-apoptotic proteins, depolarization of the mitochondrial membrane potential, and cytochrome *c* release [50]. The extrinsic apoptotic pathway is initiated by tumor necrosis factor (TNF) death ligands binding to platelets surface TNF receptors. Finally, caspase-3 activation induces phosphatidylserine (PS) exposure and platelet microparticles (PMPs) formation, generating thrombotic phenomena [51].

Otherwise, mammalian erythrocytes (red blood cells, RBCs) have been considered unable to undergo apoptosis because they contain neither mitochondria nor nucleus. However, RBCs contain procaspase-3 and procaspase-8 levels comparable with those found in Jurkat cells [52]. They can express caspase-3 and caspase-8 [53], but they do not display other elements of the apoptotic machinery, such as Apaf-1, cytochrome *c*, and caspases-2, -6, -7 and -9 [54]. Klatt et al. (2018) reported that significant receptors belonging to the tumor necrosis factors (TNF) family (CD95 [55] and Fas [56]) signaling in RBCs are known to induce a particular type of programmed cell death, similar to the apoptotic death of nucleated cells named eryptosis [57], by caspase-3 activation, leading to cell shrinkage and cell membrane scrambling [58] with PS externalization [59]. The major trigger of eryptosis is the increase of cytosolic Ca^2+^ activity resulting from Ca^2+^ entry through Ca^2+^-permeable unselective cation channels (permeable to both Na^+^ and Ca^2+^) [60]. Instead, Ca^2+^ entry and Ca^2+^-dependent RBCs membrane scrambling do not require caspases activation [61,62].

Leucocytes (white blood cells, WBCs) apoptosis displays morphological features like in other nucleated cells; however, this PCD involves distinct molecular mechanisms in various WBCs types. For instance, the extrinsic apoptosis pathway in monocytes is modulated by CD95, Fas, and TNF-cell surface apoptosis-triggering receptors (TRAIL-R1 and TRAIL-R2); it recruits cytoplasmic adaptor proteins, forming a death-inducing signaling complex (DISC) [63]. Moreover, various apoptotic agents (including commonly used chemotherapeutic drugs) induce the release of cytochrome *c* and the second mitochondria-derived activator of caspase/direct inhibitor of apoptosis-binding protein (Smac/DIABLO) in the intrinsic pathway; both proteins determine caspase-3 activation [63] in this WBCs type.

Therefore, we aim to explore cell death mechanisms in our study, analyzing the cytotoxic effects of usnic acid and *U. barbata* extracts. Usnic acid can be obtained by organic synthesis, but it can be isolated from various lichens extracts [64]. A previous report has described UA extraction from *U. barbata* acetone extract [65]; however, this present study proposes to show usnic acid isolation from UBEA. Because relatively few studies are focused exclusively on proving their effects on normal cells [66], the cytotoxicity of isolated UA and *U. barbata* dry extracts (UBA and UBE) on human blood cells cultures was evaluated in our work. Consequently, our study aims to investigate cell death mechanisms, analyzing cellular apoptosis, caspase 3/7 activity, nuclear shrinkage, lysosomal activity, ROS levels, cell cycle, and DNA synthesis by flow cytometry techniques. Finally, we suggest a relationship overview between UA, UBA, UBE concentrations, and cytotoxic activity on human blood cells cultures.

## 2. Materials and Methods

### 2.1. Lichen Samples and Usnic Acid Isolation

*U. barbata* was harvested from the Călimani Mountains (900 m altitude, Suceava County, Romania). Three dry extracts were obtained in different solvents: ethyl acetate (Chemical Company S.A., Iasi, Romania), acetone and ethanol (Chimreactiv SRL Bucharest, Romania) using a method described in detail in our previous study [13]. The dry extract in ethyl acetate was used only for usnic acid isolation. Further, in vitro studies were performed with isolated UA, UBA and UBE dissolved in 1% dimethyl sulfoxide (DMSO). Therefore, we prepared sample solutions with different concentrations: UA of 25, 50, 75, 125 μg/mL and both UBA and UBE of 75, 125, 250, 500 μg/mL.

#### 2.1.1. Usnic Acid Isolation by Semi-Preparative Chromatography

This process consists of usnic acid extraction by ultra-high performance liquid chromatography (UHPLC) with photodiode array detector (PDA), followed by collecting the separated fraction. A semi-preparative technique was adapted by our UHPLC analytical method previously validated [13]. The PerkinElmer^®^ Flexar^®^ FX-15 UHPLC system was equipped with a Flexar FX PDA-Plus photodiode array detector (PerkinElmer® Waltham, MA, USA) and a Cosmosil 5-C18-AR-2 chromatographic column with a length of 150 mm and an inner diameter of 20 mm (producer: Nacalai Tesque, Kyoto, Japan); in addition, a Gilson FC 203B fraction collector (Gilson Co, Middleton, WI, USA) was used. More detailed data can be found in the Appendix A. Working conditions consisted of flow rate = 10 mL/min, column compartment temperature = 25 °C, injection volume = 400 µL, analysis time = 18 min. The mobile phase was an isocratic system of methanol/water/glacial acetic acid (80:15:5); the detection was performed at 282 nm and 254 nm. The samples were prepared at 3 mg/mL (282 nm), and 8 mg/mL (254 nm). The retention time of the usnic acid is reported around 13 min, at a flow rate of 10 mL/min.

The usnic acid peak was collected manually between 12.5 and 13.5 min (on approximately 15% of its height, preserving the peak purity) after four successive injections (4 × 400 µL 8 mg/mL UBEA in DMSO) at 254 nm. Its identity was confirmed by comparing the retention time of the most significant peak of the sample solution with the reference one [13]. Furthermore, the solution was collected in a previously weighed vial (Appendix A). The solvent was evaporated under a nitrogen stream and the vial was placed in an oven for 30 min at 105 °C. After cooling in a desiccator to remove the last solvent traces, a yellow solid matter was obtained. Finally, the isolated usnic acid amount was calculated by subtracting the empty vial mass from the whole mass. The isolated UA was dissolved in 1% DMSO (Sigma-Aldrich Chemie GmbH, Taufkirchen, Germany) and used for in vitro analyses.

#### 2.1.2. Determination of the Purity of Isolated Usnic Acid

The purity of previously isolated UA was determined using the UHPLC-PDA analytical method [13]. First, the sample resulting from drying was weighed and brought to a final concentration of 160 µg/mL. Then, three standard usnic acid solutions were prepared simultaneously (160 µg/mL). Finally, all solutions were injected in the same sequence. The purity was calculated according to the following formula:Ps% = As/Astd × Cstd/Cs × 100,
Cstd = Mstd/d × Pstd% × 1000,
Cs = Ms/d × 1000,
where, Ps% = sample purity %, As = area of the sample, Astd = area of the standard solution, Cstd = standard solution concentration (µg/mL); Cs = sample solution concentration of the sample solution (µg/mL), Mstd = standard mass weighed (mg), d = dilution, P% std = standard purity %, Ms = sample weighed mass (mg). The previously isolated usnic acid was diluted with DMSO to a 160 µg/mL concentration. Then, this sample solution was injected into the chromatographic system according to the method described in our previous study [13]. The analyzed sequence was represented by the sample solution in DMSO, and three standard solutions of 160 µg/mL (usnic acid in DMSO) considering the average aria. Next, the precision of the area expressed in % relative standard deviation (RSD) was determined. Following the calculation, the purity value (concerning an external standard) and an RSD value = 0.66% was obtained. Finally, the identity of isolated usnic acid was certified by the retention time [13] (Appendix A).

### 2.2. In Vitro Analysis of the Biological Effects of UA, UBA, and UBE on Human Blood Cells

#### 2.2.1. Human Blood Cell Cultures

Blood samples from non-smoker healthy donor (B Rh+ blood type) were collected into heparin tubes and used throughout the experiment. The heparinized blood (1.0 mL) in 6.0 mL of Dulbecco’s phosphate buffered saline with MgCl_2_ and CaCl_2_ medium (Sigma-Aldrich Chemie GmbH, Taufkirchen, Germany) supplemented with 10% bovine fetal serum (Sigma-Aldrich, Chemie GmbH, Taufkirchen, Germany), 1% L-glutamine (Merck, KGaA, Darmstadt, Germany), and antibiotics mix solution (100 µL/mL, 10,000 U penicillin, 10 mg streptomycin, 25 µg amphotericin B per 1 mL, Sigma-Aldrich, Chemie GmbH, Taufkirchen, Germany) added in 6 wells untreated Nuncleon plates were incubated in a 37 °C incubator with 5% CO_2_. After 72 h of incubation, blood cell cultures were treated with UA, UBA and UBE dissolved in 1% DMSO. Human blood samples were treated with final concentrations of 25, 50, 75, and 125 µg/mL of UA. Higher concentrations (75, 125, 250, and 500 µg/mL) of both UBA and UBE were used to treat the human blood cell cultures. In addition, the blood cells were treated with 1% DMSO as the negative control (solvent control).

#### 2.2.2. Reagents and Equipment

Our study analyses used the flow cytometer (Attune, Acoustic focusing cytometer, Applied Biosystems, part of Life Technologies, Bedford, MA, USA). Before blood cells analysis, the flow cytometer was first set by using fluorescent beads (Attune performance tracking beads, labelling and detection, Life Technologies, Europe BV, Bleiswijk, Netherlands), with standard size (four intensity levels of beads population), and the quantity was established by enumerating cells below 1 µm [67]; 10,000 cells per sample for each analysis were gated by Forward Scatter (FSC) and Side Scatter (SSC). Flow cytometry data were collected using Attune Cytometric Software v.1.2.5, Applied Biosystems, 2010.

Annexin V-FITC/PI (Bender MedSystems GmbH, Vienna, Austria) was used to observe the apoptotic cells. Activating caspases 3/7 enzymes that determine a series of reactions triggered in response to proapoptotic signals were observed with Red Magic Methodology (MR-DEVD, Caspase-3/7 Assay Kit, Abcam, Shanghai, China). Nuclear apoptosis and lysosomal activity, dual stain with Hoechst 33,342 and acridine orange from MR-DEVD, Caspase-3/7 Assay Kit were analyzed. Total ROS level evaluation was performed using ROS Assay Kit 520 nm (Life Technologies Europe BV, Bleiswijk, The Netherlands). Propidium iodide (PI) (1.0 mg/mL, Sigma-Aldrich, Chemie GmbH, Taufkirchen, Germany) and RNase A (4 mg/mL, Promega, Madison, USA) were used in cell cycle analysis. Cell proliferation assay was performed using EdU proliferation kit, iFluor 488 (Abcam, Shanghai, China). Negative control was 1% DMSO (PanBiotech, Aidenbach, Germany).

#### 2.2.3. Apoptosis Assay

After 24 h incubation, the treated blood cells with each tested solution reported to the negative control were double-stained with Annexin V-FITC/PI. Next, blood cells were incubated in flow cytometry tubes with 2 μL Annexin V-FITC and 2 μL PI (20 μg/mL) for 30 min, at room temperature, in darkness. After incubation, 1 mL of flow cytometry staining buffer (FCB) (eBioscience^TM^, Life Technologies Europe BV, Bleiswijk, The Netherlands) was added. Viable cells, early apoptotic cells, late apoptotic cells, and necrotic cells were examined at flow cytometer, using a 488 nm excitation, green emission for Annexin V-FITC (BL1 channel), and orange emission for PI (BL2 channel).

#### 2.2.4. Caspase 3/7 Assay

After 24 h incubation, 300 μL of blood cell culture was transferred to flow cytometry tubes, 20 μL of MR-DEVD solution was added and mixed with the cells. Next, 20 μL of PI was added. After incubation, was added 1 mL FCB. The early stages of cell apoptosis by activating caspase 3/7 were analyzed by flow cytometry, using a 488 nm excitation, red emission for MR-DVD (BL3 channel), and orange emission for PI (BL2 channel).2.2.5. Nuclear Condensation and Lysosomal Activity Assay

After 24 h of treatment with the tested solutions, 300 μL of blood cell culture was introduced in flow cytometry tubes, 2 μL of Hoechst 33,342 stain was added, and blood cells were mixed well. After this process, 50 μL of acridine orange (AO) 1.0 μM was added, and the cells were incubated 30 min at room temperature into darkness. After incubation, 1 mL FCB was added; the cells were examined at flow cytometer. UV excitation and blue emission for Hoechst 33,342 (VL2) at 488 nm, and green emission acridine orange (BL1 channel) were used for examination.

#### 2.2.5. Total ROS Activity Assay

After 24 h treatments with the tested solutions, 100 μL of ROS Assay Stain solution was added for each 1 mL of blood cell culture in flow cytometry tubes and mixed well. Next, the cells were incubated for 60 min at 37 °C, in an incubator with 5% CO_2_. After incubation, the blood cells were analyzed by flow cytometry, using a 488 nm excitation and green emission for ROS (BL1 channel).

#### 2.2.6. Cell Cycle Analysis

Blood cells were treated with UA (25–125 μg/mL), UBA and UBE (75–500 μg/mL) and incubated for 24 h; 1 mL of each cell culture was washed in FCB, introduced in flow cytometry tubes, and fixed with 50 μL ethanol for 10 min. After this process, the cells were treated with PI (20 μg/mL) and RNase A (30 μg/mL) and incubated for 30 min at room temperature, into darkness. After this time, 1 mL FCB was added, and the cell cycle distribution was detected at flow cytometer, using a 488 nm excitation and orange emission for PI (BL2 channel).

#### 2.2.7. Cell Proliferation Assay

After 24 h of treatment, 1 mL of blood cell culture was incubated with 50 µM EdU (500 µL), at 37 °C, for 2 h. Then, the cells were fixed (100 µL of 4% paraformaldehyde in PBS) and permeabilized (100 µL of Triton X-100 1×). After washing in 3% BSA in flow cytometry (2 mL) and centrifuging at 300× *g* for 5 min, at 4 °C, the blood cells were incubated with a reaction mix (500 µL), 30 min at room temperature, into darkness. After washing in permeabilization buffer (2 mL) and centrifuging (300× *g*, 5 min, at 4 °C), 1 mL FCB was added. Finally, the blood cells were examined by flow cytometry, using a 488 nm excitation and green emission for EdU-iFluor 488 (BL1).

#### 2.2.8. Statistical Analysis

All analyses were performed in triplicate, and the obtained results were presented as means values ± standard deviation (SD). Our results are presented as percent (%) of cell and nuclear apoptosis, caspase 3/7 activity, autophagy, cell cycle, DNA synthesis, and count (×10^4^) of oxidative cellular stress after flow cytometry analyses were performed with SPSS v. 23 software, IBM, 2015. The Levene test was analyzed for homogeneity of variances of samples, while paired *t*-test, ANOVA [68], was used to establish the differences between samples and controls, and *p* < 0.05 was considered statistically significant. Figures 2, 4, 6, 7, 9, 11 and 13 were made with Attune Cytometric Software v.1.2.5, Applied Biosystems, 2010 (Bedford, MA, USA). Figures 3, 5, 8, 10, 12 and 14 were made by the v. 14.8.1, 2014 of MedCalc program (Ostend, Belgium). 

## 3. Results

### 3.1. Usnea Barbata Dry Extracts and Usnic Acid Isolation

The obtained chromatograms in both UHPLC determinations from Section 2.1.1 and Section 2.1.2 are presented in Figure 1. From 12.8 mg UBEA, 3.6 mg of isolated usnic acid (Figure 1d) with 89.36% purity was obtained. The yield of this process was 28.15%.

### 3.2. In Vitro Analysis of the Biological Effects of UA, UBA, and UBE on Human Blood Cells

#### 3.2.1. Cell Apoptosis Assay

Cell apoptosis induced by UA, UBA, and UBE treatments was determined based on morphology and cell membrane integrity in blood cell cultures. The obtained results are illustrated in Figure 2A–C and Figure 3a–c (V-cell viability, EA-early apoptosis, LA-late apoptosis, N-necrosis).

The influence of UA (25, 50, 75, 125 µg/mL) on blood cells viability and apoptosis is presented in Figure 2A(b–e) and Figure 3a.

It can be noted that the viability of blood cells treated with 25 µg/mL of UA (Figure 2A(a,b)) insignificantly decreased in comparison with the solvent control: 96.45 ± 0.27% vs. 96.89 ± 0.14% (*p* ≥ 0.05, Figure 3a). On the other hand, a concentration of 50 µg/mL of UA on blood cell cultures (Figure 2A(a,c)) determined reduced cell viability reported to 1% DMSO: 95.75 ± 0.63% vs. 96.89 ±0.14% (*p* < 0.05, Figure 3a).

Likewise, these low concentrations of UA (25 and 50 µg/mL) induce insignificant differences of early apoptosis (Figure 2A(a–c)) collated to control: 3.12 ± 0.26%; 3.69 ± 0.71% vs. 2.72 ± 0.16% (*p* ≥ 0.05, Figure 3a).

Moreover, higher concentrations of UA (75 and 125 µg/mL) significantly influenced both parameters (Figure 2A(a,d,e)). They induced an evident decline of cell viability (71.34 ± 0.90%; 61.43 ± 0.88% vs. 96.89 ±0.14%, *p* < 0.001), and an augmentation of early apoptosis (27.27 ± 1.00%; 37.04 ± 0.66% vs. 2.72 ± 0.16%, *p* < 0.001, Figure 3a).

UBA activity (75, 125, 250, 500 µg/mL) on blood cells viability and apoptosis compared with 1% DMSO is shown in Figure 2B(a–e) and Figure 3b.

The obtained data revealed that 75 µg/mL of UBA (Figure 2B(a,b)) determined a diminution in cell viability: 80.16 ± 0.57% vs. 96.89 ± 0.14%, (*p* < 0.001); also, it induced an increase of cell apoptosis: 19.45 ± 0.60% vs. 2.72 ± 0.16% (*p* < 0.001, Figure 3b).

Higher concentrations of UBA (125, 250, and 500 µg/mL) remarkably reduced blood cells viability, triggering apoptosis (Figure 2B(c–e)). Hence, previously mentioned concentrations of UBA had a significant cytotoxic effect on blood cells, with diminishing viability compared with solvent control: 66.93 ± 1.37%; 54.57 ± 0.65%; 52.15 ± 0.81%; vs. 96.89 ± 0.14% (*p* < 0.001, Figure 3b). Moreover, these results indicated high rise of early apoptosis: 32.18 ± 1.22%; 43.99 ± 0.66%; 45.98 ± 0.78% vs. 2.72 ± 0.16% (*p* < 0.001, Figure 3b).

The flow cytometry results regarding UBE effects on the apoptosis process are indicated in Figure 2C(b–e) and Figure 3c.

It can be seen that 75 µg/mL of UBE (Figure 2C(a,b)) determined a diminution of blood cell viability reported to 1% DMSO: 86.66 ± 0.45% vs. 96.89 ± 0.14%, *p* < 0.001; therefore, it raised their apoptosis: 12.81 ± 0.66% vs. 2.72 ± 0.16% (*p* < 0.001, Figure 3c).

In addition, higher concentrations of UBE (125, 250, and 500 µg/mL) had a considerable cytotoxic effect on blood cells (Figure 2C(a,c–e)); viability has substantial reduced values, compared with the solvent control: 65.96 ± 0.68%; 57.91 ± 0.96%; 42.65 ± 0.32% vs. 96.89 ± 0.14% (*p* < 0.001, Figure 3c). In addition, our results indicate that UBE at the same concentrations promoted significantly augmented levels of early apoptosis reported to 1% DMSO: 30.19 ± 0.77%; 30.99 ± 0.77%; 45.52 ± 0.18% vs. 2.72 ± 0.16% (*p* < 0.001, Figure 3c).

Finally, Figure 2 indicates that insignificant late apoptosis and necrosis phenomena occurred in blood cell cultures after 24 h treatment.

#### 3.2.2. Caspase 3/7 Activity Assay

The apoptotic effects of UA, UBA, and UBE evaluated by measuring the caspase-3/7 activity compared with 1% DMSO, were registered in Figure 4A–C and Figure 5a–c.

We noted that the minimum concentration of UA (25 µg/mL) induces a low increase of cell apoptosis (Figure 4A(a,b)) reported to control (3.75 ± 0.36% vs. 1.38 ± 0.03%, *p* < 0.01, Figure 5a). Forwards, a remarkable increase of caspase-3/7 activation was registered on blood cell cultures treated with 50, 75, and 125 µg/mL of UA (Figure 4A(a,c–e)) in comparison with 1% DMSO: 6.81 ± 0.43%; 29.49 ± 1.96%; 44.74 ± 0.41% vs. 1.38 ± 0.03% (*p* < 0.001, Figure 5a).

Remarkably, the lowest UBA concentration (75 µg/mL, Figure 4B(a,b)) induces mild apoptosis in blood cells cultures: 16.16 ± 0.93% vs. 1.38 ± 0.03% (*p* < 0.01, Figure 5b). Furthermore, we aimed to confirm that 125, 250, and 500 µg/mL of UBA produce significant blood cells apoptosis (Figure 4B(c–e)). Consequently, we evaluated the intracellular activity of effector caspase 3/7, and we observed that the biochemical cascade of reactions implied into pro-apoptotic signal has considerably increased more than 1% DMSO: 22.35 ± 1.58%; 32.53 ± 0.57%; 43.57 ± 0.73% vs. 1.38 ± 0.03%, (*p* < 0.001, Figure 5b).

Similarly, 75 µg/mL of UBE (Figure 4C(a,b)) induced a low apoptosis in blood cells cultures: 11.25 ± 0.96% vs. 1.38 ± 0.03% (*p* < 0.01, Figure 5c). Higher UBE concentrations (125, 250, and 500 µg/mL) triggered proapoptotic signal with considerable increased values (Figure 4C(a,c–e)) compared with solvent control: 18.15 ± 0.52%; 30.18 ± 0.09%; 43.54 ± 0.72% vs. 1.38 ± 0.03% (*p* < 0.001, Figure 5c).

Our results indicate a similar trend to the previous assay in UA, UBA, and UBE activity on blood cell cultures (Figure 2). The effect on caspase 3/7 activity is directly proportional with the sample concentration, which registers significantly increased levels at high doses, and decreases in the order of: usnic acid, *U. barbata* dry extract in acetone, and ethanol (Figure 4 and Figure 5).

#### 3.2.3. Nuclear Condensation and Lysosomal Activity Assay

Apoptosis is the mode of cell death that includes pyknosis; in this assay, pyknotic nuclei were stained with Hoechst 33,342 on blood cell cultures. Another aimed objective was an evaluation of lysosomal activity directly related to autophagy.

Therefore, blood cells were also colored with acridine orange (AO). The obtained results were synthesized in Figure 6 and Figure 7.

The lowest concentration of UA (25 µg/mL, Figure 6A(a,b)) induced an insignificant increase of nuclear condensation: 1.36 ± 0.20% vs. 1.03 ± 0.03% (*p* ≥ 0.05, Figure 8a) and a mild increase of lysosomal activity (Figure 7A(a,b)) reported to solvent control: 6.59 ± 0.33% vs. 1.04 ± 0.04% (*p* ≤ 0.01, Figure 8a).

The higher concentrations of UA (50, 75, and 125 µg/mL, Figure 6A(a,c–e)) continued to have directly proportional effects on nuclear shrinkage: 1.49 ± 0.02%; 3.00 ± 0.10%; 3.19 ± 0.30% vs. 1.03 ± 0.03% (*p* ≤ 0.01; *p* < 0.001, Figure 8a). In addition, the same UA concentrations (Figure 7A(a,c–e)) induced a substantial increase of the autophagy levels compared with 1% DMSO: 12.97 ± 1.55%; 21.72 ± 0.38%; 27.05 ± 1.52% vs. 1.04 ± 0.04% (*p* ≤ 0.01; *p* < 0.001, Figure 8a).

Nuclear shrinkage and autophagy were concomitantly examined to evaluate the mechanism of UBA cytotoxicity on blood cells cultures. Thereby, it can be noted that 75 µg/mL of UBA had minimal effects on both processes (Figure 6B(a,b) and Figure 7B(a,b)) compared with 1% DMSO: nuclear condensation 1.41 ± 0.09% vs. 1.03 ± 0.03%, (*p* < 0.05, Figure 8b), and autophagy 4.64 ± 0.38% vs. 1.04 ± 0.04% (*p* ≤ 0.01, Figure 8b).

On nuclear condensation, the following concentrations of UBA: 125, 250, and 500 µg/mL (Figure 6B(a,c–e)) continued to show mild effects reported to control: 1.85 ± 0.10%; 2.36 ± 0.16%; 4.41 ± 0.32% vs. 1.03 ± 0.03% (*p* < 0.01, *p* ≤ 0.001, Figure 8b). In addition, previously mentioned concentrations of UBA (Figure 7B(a,c–e)) significantly increased autophagy: 13.49 ± 0.45%; 21.99 ± 0.57%; 42.32 ± 0.85% vs. 1.04 ± 0.04% (*p* ≤ 0.001, Figure 8b).

Finally, 75 µg/mL UBE (Figure 6C(a,b)) showed a similar effect on nuclear contraction with UBA (Figure 7B(a,b)) at the same concentration as the solvent control: 1.85 ± 0.03% vs. 1.03 ± 0.03% (*p* ≤ 0.01, Figure 8c).

However, this effect considerably increases at the following UBE higher concentrations (125, 250 and 500 µg/mL, Figure 6C(a,c–e)) reported to control: 4.29 ± 0.06%; 14.27 ± 0.93%; 18.64 ± 1.22% vs. 1.03 ± 0.03% (*p* ≤ 0.001, Figure 8c)

Besides, UBE acted slowly, inducing a moderate increase of the lysosomal activity (Figure 7C(a–e)) from 75 µg/mL to 500 µg/mL reported to 1% DMSO: 2.87 ± 0.09%; 5.59 ± 0.44%; 11.08 ± 1.21%; 16.77 ± 0.69% vs. 1.04 ± 0.04% (*p* ≤ 0.01, *p* ≤ 0.001, Figure 8c).

#### 3.2.4. Total ROS Activity Assay

The ranges 25–125 µg/mL UA and 75–500 µg/mL UBA and UBE were selected to evaluate oxidative stress in blood cells by ROS level determination.

As shown in Figure 9A–C(a–e), except for 1% DMSO, all samples (UA, UBA, and UBE) induced ROS generation, highlighted by the moving of the peaks to the right of the graph. Hence, the lowest concentration of UA (25 µg/mL) slowly stimulated ROS production (Figure 9A(a,b)) reported to 1% DMSO: 34.33 × 10^4^ ± 4.04 vs. 10.40 × 10^4^ ±1.00 (*p* < 0.01, Figure 10a). A remarkable increase in ROS levels was observed in blood cells treated with 50, 75, and 125 µg/mL of UA (Figure 9A(a,c–e)) compared with the negative control: 56.33 × 10^4^ ± 1.52; 80.33 × 10^4^ ± 0.57; 84.67 × 10^4^ ± 0.57 vs. 10.40 × 10^4^ ± 1.00 (*p* < 0.001, Figure 10a).

The lowest concentration of UBA (75 µg/mL) slightly stimulated ROS production (Figure 9B(a,b)) compared with 1% DMSO: 21.00 × 10^4^ ± 1.00 vs. 10.40 × 10^4^ ± 1.00 (*p* < 0.001, Figure 10b).

Subsequent higher concentrations of UBA (125, 250, and 500 µg/mL) continued to increase ROS levels (Figure 9B(a,c–e)) and the differences reported to solvent control remained significant: 26.00 × 10^4^ ± 1.00; 35.66 × 10^4^ ± 1.15; 63.66 × 10^4^ ± 3.21 vs. 10.40 × 10^4^ ± 1.00 (*p* < 0.001, Figure 10b).

Likewise, 75, 125, 250, and 500 µg/mL UBE considerably stimulated ROS production in blood cells, directly proportional with the concentrations (Figure 9C(a–d)).

Our results showed that ROS levels in blood cells compared with the negative control were as follows: 24.66 × 10^4^ ± 0.57; 35.63 × 10^4^ ± 0.57; 46.00 × 10^4^ ± 1.00; 62.53 × 10^4^ ± 2.50 vs. 10.40 × 10^4^ ± 1.00 (*p* < 0.001, Figure 10c).

#### 3.2.5. Cell Cycle Analysis

To explore the effects of UA, UBA, and UBE on cell cycle distribution on blood cell cultures, the DNA content was evaluated. Propidium iodide/RNase A staining was performed using flow cytometry analyses for DNA content (Figure 11A–C).

As shown in Figure 11A(a–e) and Figure 12a, UA concentrations of 25, 50, 75, and 125 µg/mL induce a noteworthy cell cycle arrest in the G1/G0 phase: 64.13 ± 1.55%; 78.52 ± 0.87%; 81.91 ± 1.41%; 88.09 ± 0.98% vs. 39.29 ± 0.76%; *p* < 0.01, *p* < 0.001 compared to solvent control. This activity is directly proportional to UA concentrations.

To understand whether the cell growth inhibition was due to cell cycle arrest, blood cells were treated with UBA of 75–500 µg/mL concentrations (Figure 11B(a–e)). *U. barbata* dry extracts exhibited a noticeable cell cycle arrest in G0/G1 phase reported to 1% DMSO: 65.13 ± 0.15%; 76.35 ± 0.94%; 78.93 ± 0.54%; 81.86 ± 1.11%; vs. 39.29 ± 0.76% (*p* < 0.01, *p* < 0.001, Figure 12b).

Thereby, 75, 125, 250, and 500 µg/mL of UBE (Figure 11C(a–e)) induced cell cycle arrest in G0/G1 phase reported to the negative control as follows: 68.16 ± 0.14%; 68.47 ± 0.58%; 76.06 ± 0.68%; 82.75 ± 0.55% vs. 39.29 ± 0.76% (*p* < 0.001, Figure 12c).

Finally, it can be observed that UA proved the highest effect on cell cycle arrest in G0/G1, followed by UBA and UBE with similar activities.

#### 3.2.6. Cell Proliferation Assay

Flow cytometry analyses with EdU incorporation were used for examining DNA synthesis in blood cells (Figure 13).

In this study, we observed that the lowest concentration (25 µg/mL) of UA determined an increase of DNA synthesis (Figure 13A(a,b)) in comparison with 1% DMSO (17.25 ± 0.36% vs. 11.43 ± 1.04%, *p* < 0.05, Figure 14a).

Instead, UA at 50, 75 and 125 µg/mL did not significantly alter DNA synthesis (Figure 13A(a,c–e)) relative to the control (11.25 ± 0.83%; 10.32 ± 0.64%; 6.49 ± 1.25% vs. 11.43 ± 1.04%, *p* ≥ 0.05, Figure 14a).

The lowest concentration of UBA (75 µg/mL) did not significantly modify the DNA synthesis (Figure 13B(a,b)) more than 1% DMSO: 12.78 ± 0.67% vs. 11.43 ± 1.04% (*p* ≥ 0.05, Figure 14b).

Regarding higher concentrations of UBA (125 and 250 µg/mL), a decrease of DNA synthesis (Figure 13B(a,c,d)) was registered, compared with 1% DMSO, with moderate differences: 3.12 ± 0.18%; 4.81 ± 0.15% vs. 11.43 ± 1.04% (*p* < 0.05, Figure 14b).

Finally, the highest UBA concentration (500 µg/mL) did not significantly affect DNA synthesis (Figure 13B(a,e)) more than the solvent control: 10.77 ± 0.43% vs. 11.43 ± 1.04% (*p* ≥ 0.05, Figure 14b).

In blood cells treated with 75 µg/mL UBE, an evident higher stimulation of DNA synthesis reported to solvent control (19.05 ± 0.64% vs. 11.43 ± 1.04%, *p* < 0.01) was registered (Figure 13C(a,b) and Figure 14c).

The treatment with 125 and 250 µg/mL of UBE (Figure 13a,d) did not significantly alter DNA synthesis in comparison with 1% DMSO: 9.92 ± 0.43%; 10.60 ± 0.63% vs. 11.43 ± 1.04%, (*p* ≥ 0.05, Figure 14c).

Furthermore, 500 µg/mL of UBE produced a lower DNA synthesis stimulation (Figure 13C(a,e)) than the negative control: 8.89 ± 0.30% vs. 11.43 ± 1.04% (*p* < 0.05, Figure 14c).

## 4. Discussion

Our previous study analyzed five *U. barbata* dry extracts in different solvents [13]; we calculated the yield and evaluated the usnic acid, total polyphenols, and tannins content of each obtained extract. Therefore, we opted only for three lichen dry extracts for the present study: in ethyl acetate, acetone, and ethanol.

The highest usnic acid content (376.73 µg/mg) was the reason for selecting UBEA in the first phase of our study-usnic acid isolation. Then, isolated usnic acid was purified in the sample matrix. As a result, we obtained usnic acid with 89.36% purity and a yield of 28.15%. Ranković et al. (2012) obtained 95 mg usnic acid with 98.6% purity from 500 mg *U. barbata* dry acetone extract [59]. The purity value difference could be due to the solvents used. We used only DMSO, while in the previously mentioned study, dry acetone extract was dissolved in benzene and then, usnic acid was recrystallized using chloroform/ethanol. We can state that our isolated usnic acid is (+)-UA because *Usnea* sp. tends to produce this enantiomer exclusively; in addition, (+)-UA registers antiviral, insecticidal, and phytotoxic activities significantly higher than (−)-UA [69].

Isolated UA and both UBA and UBE were used to evaluate their cytotoxic activity on blood cells. We opted for UBA because it contains an appreciable usnic acid amount (282.78 µg/mg) and other secondary metabolites. In addition, only 127.21 µg/mg of UA were extracted in UBE and a wider lichen secondary metabolites variety. UBE also had the highest extraction yield (12.52%) compared with UBA (6.36%) and UBEA (6.27%) [13]. Consequently, the obtained biological effects could be correlated with the secondary metabolites content.

The present study proved that UA generally induced a significant cytotoxic effect on normal blood cells, more intense than both *U. barbata* extracts, UBA, and UBE. Hence, in early apoptosis events, the appearance of PS residues (commonly hidden within the plasma membrane) on the surface of the cells can be used to detect and measure this PCD. In our flow cytometry method, we opted for annexin V as staining to detect apoptotic cells due to its ability of PS-binding [70]. Moreover, translocation of PS to the external cell surface can occur during apoptosis and necrosis. The difference between these two forms of cell death is that the cell membrane remains intact in early apoptosis; however, it loses integrity and becomes permeable when necrosis is installed [71]. Shlomovitz et al. (2019) showed that PS externalization is also available in necroptosis [72]. However, in RBCs, this process corresponds to eryptosis (quasi-apoptosis) [73]. The intact cells membrane consists of the bilayer with choline-containing phospholipids (phosphatidylcholine and sphingomyelin) in the outer layer and amine-containing phospholipids (phosphatidylethanolamine and PS) in the inner layer. This normal disposition is known as phospholipid asymmetry. Lipid asymmetry is disturbed when the erythrocytes enter into eryptosis, and PS is exposed on RBCs surface [74]. This process involves three ATP and Ca^2+^ -dependent transporters activation (flippase [75], floppase [76], and scramblase [77]). In addition, spectrin [78,79] oxidation and increasing cytoplasmic Ca^2+^ [80] concentration lead to membrane proteins denaturation [81]. Usnic acid oxidative stress-induced plays an essential role in all blood cells PCD, triggering these various molecular mechanisms [35]. One of the prominent protein families that regulate and execute programmed cell death is caspases; they cleave a subset of essential cellular proteins to promote apoptotic cell death [82].

In all blood cells types PCD involves caspases [50,52,53,59,63,73,83,84,85,86]. In response to apoptotic stress, the activated initiator caspases (caspase-2, 8, 9) cleave and activate the effector caspases (caspase-3, 6, 7), which execute the death process [87]. They regulate the extrinsic (receptor-mediated) apoptosis pathway involving receptor binding, followed by activation of the initiator caspase, caspase-8, which activates caspase-3 or amplifies caspase-3 activation cleaving BH3 Interacting Domain Death Agonist protein. Bid cleavage induces mitochondrial cytochrome *c* release, forming a protein complex (apoptosome) and activating caspase-9 [88]. During apoptosis, caspase-3 is also actively transported to the nucleus through the nuclear pores, playing a significant role in its disintegration by processing several nuclear substrates. Caspase-7 plays a significant role in cell viability loss [89]. According to Sundquist et al. (2006), the late apoptotic events occur after activating the effector caspases. Late apoptosis includes exposure of phosphatidylserine on the external surface of the plasma membrane (which can be measured by annexin V binding), cleavage of poly (ADP-ribose) polymerase (PARP), and internucleosomal DNA fragmentation [48]. McComb et al. (2019) revealed that efficient apoptosis requires feedback amplification of upstream apoptotic signals by effector caspase-3 or -7 [90]. For this reason, we aimed to evaluate the influence of our tested samples on caspase 3/7 apoptosis pathway. The obtained results showed that caspase 3/7 activity was significantly stimulated during PCD process.

During apoptosis, caspase-3 is actively transported to the nucleus through the nuclear pores, playing a significant role in its disintegration by processing several nuclear substrates [89]. Nuclear apoptosis is characterized by chromatin condensation and progressive DNA cleavage into high-molecular-weight fragments and oligo-nucleosomes [91]. We analyzed the nuclear shrinkage to validate that UA, UBA, and UBE caused apoptosis; this process occurs only in white blood cells [92], RBCs, and platelets being enucleate. Chromatin condensation and fragmentation of nuclei are included in PCD [93]; exclusively, UA at high concentrations showed an appreciable stimulatory effect compared with the solvent control and UBA, and UBE samples.

Various studies have recently shown that lysosomes have been implicated in the regulation of cell death. Increasing their membrane permeability, released hydrolytic enzymes can contact cytosolic targets and contribute to apoptotic cell death [94]. Furthermore, lysosomal activity is directly associated with autophagy, another decisive process for cell death [95]. Various studies from accessed scientific literature analyze autophagy in platelets [96,97] and different WBCs [98,99,100].

Moreover, according to numerous studies, mitochondrial dysfunction consists of structural alteration, membrane potential disruption, and electron transport reaction instability. These events generate ROS overproduction, caspase cascades activation, and apoptosis pathway initiation [101]. The mitochondrial apoptotic pathway is available only at platelets [102,103], and WBCs [104].

In addition, low ROS concentration can promote cell proliferation, whereas excessive ROS levels cause DNA oxidative damage, consequently inducing cell death [105]. ROS levels are implicated in all blood cell lines death: platelets [106,107], RBCs [108,109], and WBCs [110,111].

It is known that the cell cycle consists of few successive phases in mammals: synthesis (S) with DNA replication and mitosis (M) with repartition of replicated DNA into two daughter cells. Separation of DNA replication from mitosis is performed by two gap phases (G1 and G2). In G1, the cell increasing size, RNA, and protein synthesis occurs, while in G2, after DNA synthesis, the cell grows and synthesizes proteins. Only in RBCs cell cycle arrest occurs in G1 [112]. After division, the cell enters the resting phase, known as the G0 gap phase [113]. Thus, cell cycle arrest represents the first response to DNA damage and one of the first steps in cell apoptosis. However, nucleate WBCs proliferation is a tightly controlled process, and DNA replication is essential [114].

According to the obtained results, the high ROS levels produced by 75 and 125 µg/mL of UA in blood cells triggers a series of consecutive events: cell apoptosis, effector caspases 3/7 activation, pyknosis, autophagy, and cell-cycle arrest in the G0/G1 phase. Furthermore, many studies on numerous cell types proved apoptosis induction by ROS [115].

We have also shown that 125, 250, and 500 µg/mL concentrations of UBA and UBE triggered blood cells apoptosis by caspase 3/7 pathway, oxidative stress, and accumulation of cells in G0/G1. Moreover, altered cell-cycle checkpoints and cell apoptosis parameters conformed with references [116].

Our results proved that UA and UBE at the lowest concentrations stimulated DNA replication (S-phase of the cell cycle). At higher concentrations, they highlighted inhibitory activity on blood cell proliferation. In great measure, these opposite effects manifested at low and high concentrations and can be associated with ROS levels, as previously mentioned. Similar data are mentioned by Damiano et al. (2019), revealing ROS dual function in skeletal muscle: at low levels, they improve muscle force and adaptation to exercise, while at high levels, they decrease muscle performance [117].

However, relatively few studies from the accessed scientific literature are focused exclusively on proving the effects of *Usnea* sp. extracts and usnic acid on normal cells [60,118]. A few years ago, dietary supplements containing usnic acid used for weight loss (Lipokinetix, [119]) reported severe hepatotoxic effects [120,121,122]. Fujimoto et al. (2010) reported that in vitro usnic acid hepatotoxicity involves oxidative stress, mitochondrial toxicant depletion of glycogen [123], and potential ATP biosynthesis inhibition mediated by mitochondrial electron transport chain [124], triggering necrotic death of hepatocytes. The structural properties of usnic acid can explain these molecular mechanisms. Thus, usnic acid is a lipophilic compound that can easily pass the mitochondrial membranes into the matrix, releasing a proton. Next, usniate anion diffuses into the intermembrane space to bind to a proton to restore usnic acid. The resulting cycle causes a proton leak that could dissipate the proton grad across the membrane, altering ATP levels and changing mitochondrial membrane potential [125]. Usnic acid can induce structural changes in intracellular glutathione molecules, decreasing its reduced form (GSH) [126]. In the same mode, usnic acid can perform spectrin oxidation with cell membrane shrinkage and PS exposure during apoptosis. Thus, usnic acid prooxidant potential can induce oxidative stress and liver cell death signaling. Concomitantly, using natural or synthetic antioxidants to neutralize the prooxidative activity of UA might also be a cell-protecting measure. Due to high hepatotoxicity induced by usnic acid, FDA released a Safety Alerts for Human Medical Products about Lipokinetix [119]. This notification is also current and supports the necessity for identification of natural or synthetic compounds to ensure the safe use of AU as potential anticancer drug.

Instead, the most numerous researchers pointed out these activities on various tumor cell lines [127]. For instance, Ozturk et al. (2019) reported several extract types of different *Usnea* sp. which determine cell apoptosis and DNA damage on cancer cells [128]. In addition, methanol extracts of *U. barbata* induced cell apoptosis, as evidenced by the increasing Annexin V expression and pan-caspase activation in human breast and lung cancer cells [16]. Additionally, Disoma et al. (2018) mentioned caspase 3/7 activation as an apoptotic mechanism on colon cancer cells proved by *U. filipendula* extracts [129]. In their study, Koparal et al. (2006) examined usnic acid effects on two types of lung cells (normal and tumor cells). They described usnic acid cytotoxic activity on both cell types and highlighted that cancer cells are more sensitive than normal cells [130]. Geng et al. (2018) reported that usnic acid induces cycle arrest, apoptosis, and autophagy in gastric cancer cells types in vitro and in vivo [131]. In another study, Wang et al. (2019) proved antileukemia action of (+) usnic acid derivatives, inhibiting pan-Pim kinases [116]. Besides, Rabelo et al. (2012) suggested that usnic acid displays variable redox-active properties, acting as an antioxidant and prooxidant agent, according to different system conditions and cellular environment [27]. High cytotoxic effects on cancer cells and minimal unwanted effects on normal cells represent the essential quality of an antitumor agent. High cytotoxicity levels on cancer cells and low damage on normal cells represent the meaningful purpose in antitumor activity. Finally, Tram et al. (2020) evidenced the highly different cytotoxicity of the same compound against tumor and normal cells [23].

Our study highlights the relationship between concentration and biological effect on normal blood cells. Thereby, usnic acid at a minimal concentration (25 µg/mL) shows low cytotoxicity on human blood cells, slowly inducing cell apoptosis, caspases 3/7 activation, mild ROS level and stimulating DNA synthesis. On the other hand, higher concentrations (50–125 µg/mL) of UA progressively display significant cytotoxic effects: increasing cell apoptosis, effector caspases 3/7 proapoptotic signal, nuclear condensation, autophagy, oxidative stress, and causing cell-cycle arrest in G1/G0 phase.

*U. barbata* dry extracts in acetone and ethanol, at low concentration (75 µg/mL), exhibit minor cytotoxicity, inducing cell and nuclear apoptosis, autophagy, and increased DNA synthesis. In contrast, higher concentrations (125–500 µg/mL) of UBA and UBE report directly proportional significant toxic effects on blood cells, enhancing cell and nuclear apoptosis, autophagy, ROS levels and promoting cell-cycle arrest G1/G0 phase.

## 5. Conclusions

The novelty of our study consists of analyzing *U. barbata* and usnic acid cytotoxic effects on human normal blood cells cultures. The principal points of this complex activity have been highlighted, exploring the cell and nuclear apoptosis, caspase 3/7 activity, autophagy, oxidative cellular stress, cell cycle, and DNA synthesis.

High cytotoxicity levels on cancer cells and relatively lower damage to the normal blood cells represent the meaningful purpose in antitumor activity. Based on this statement, our study can offer essential data to target the previously mentioned objective by evaluating the blood cells sensitivity to various concentrations of *U. barbata* dry extracts and usnic acid. The obtained results suggest that future researches may select several concentration domains to evaluate their various pharmacological activities. Moreover, exploring anticancer potential, we can select which extracts highlight optimal and exclusive cytotoxicity on a broad domain of cancer cells, also displaying minimal or no side effects on normal cells.

## Figures and Tables

**Figure 1 antioxidants-10-01171-f001:**
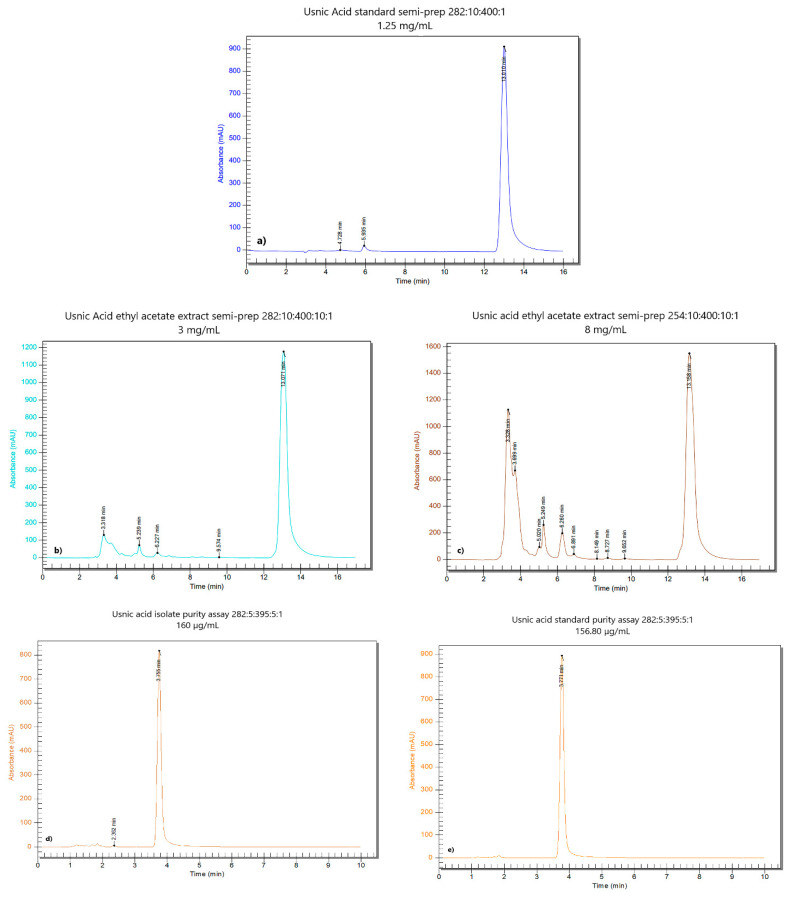
The chromatograms of usnic acid and sample solution for usnic acid separation; (**a**–**c**) chromatograms of usnic acid from semi-preparative chromatography, at the same flow rate (10 mL/min) and different wavelengths and concentrations: (**a**) usnic acid standard at 282 nm and 1.25 mg/mL; (**b**) usnic acid in UBEA at 282 nm and 3 mg/mL; (**c**) usnic acid in UBEA at 254 nm and 8 mg/mL; (**d**,**e**) chromatograms obtained at 282 nm from purity determination of isolated usnic acid: (**d**) sample solution (usnic acid isolate) (**e**) usnic acid standard solution.

**Figure 2 antioxidants-10-01171-f002:**
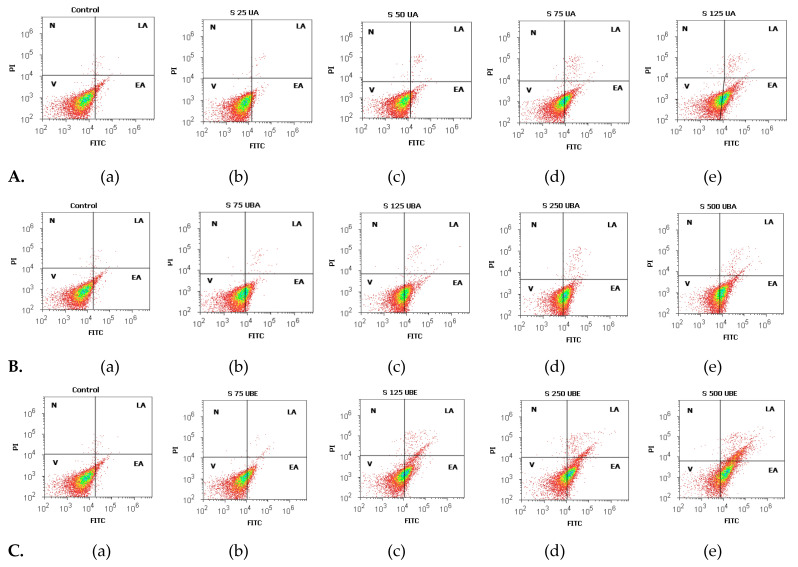
Cell apoptosis models of usnic acid (UA), U. barbata acetone UBA, and U. barbata ethanol (UBE) treatments in normal blood cell cultures. Annexin V-FITC/PI patterns of 1% dimethyl sulfoxide (DMSO) Negative Control (**A**(**a**), **B**(**a**), **C**(**a**)); **A**(**b**–**e**) UA 25, 50, 75, 125 µg/mL; **B**(**b**–**e**) UBA 75, 125, 250, 500 µg/mL; and **C**(**b**–**e**) UBE 75, 125, 250, 500 µg/mL.

**Figure 3 antioxidants-10-01171-f003:**
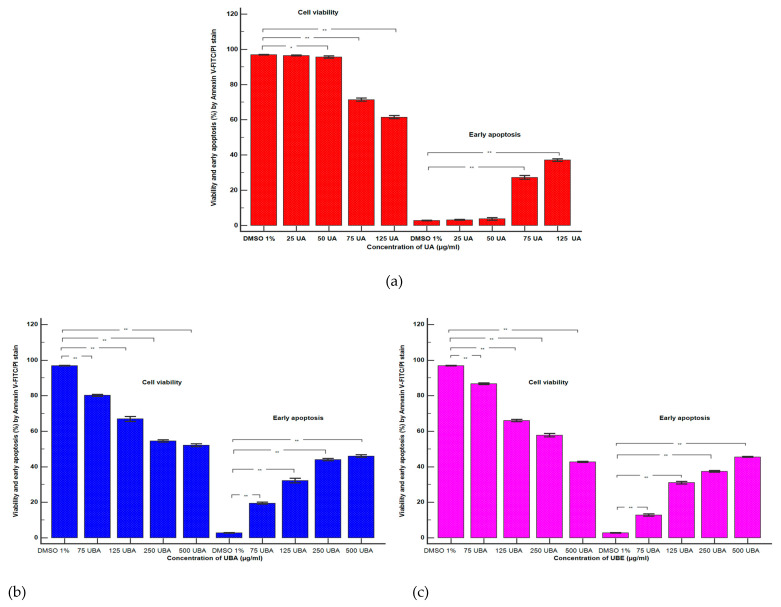
Statistical analysis of cell apoptosis: (**a**) UA; (**b**) UBA; (**c**) UBE. * *p* < 0.05 and ** *p* ≤ 0.001 represent significant statistical differences between control and samples made by paired samples *t*-test.

**Figure 4 antioxidants-10-01171-f004:**
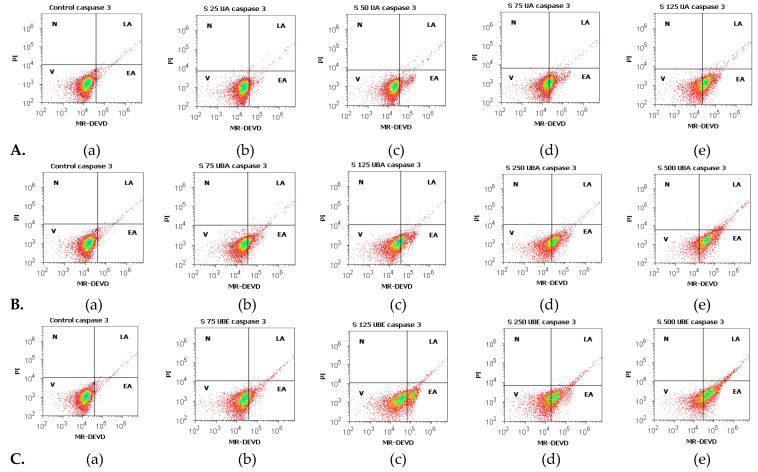
Caspase 3/7 activity status of UA, UBA, and UBE treatments in normal blood cell cultures. MR-DEVD/PI patterns of 1% DMSO as Negative Control (**A**(**a**), **B**(**a**), **C**(**a**)); **A**(**b**–**e**) UA 25, 50, 75, 125 µg/mL; **B**(**b**–**e**) UBA 75, 125, 250, 500 µg/mL; **C**(**b**–**e**) UBE 75, 125, 250, 500 µg/mL.

**Figure 5 antioxidants-10-01171-f005:**
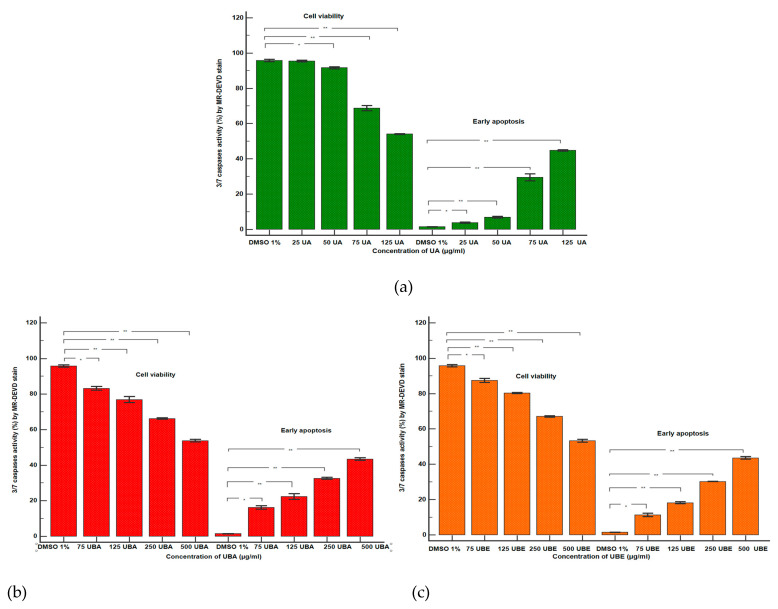
Statistical analysis of caspase 3/7 activity: (**a**) UA; (**b**) UBA; (**c**) UBE. * *p* < 0.01 and ** *p* < 0.001 represent significant statistical differences between the control and samples made by paired samples *t*-test.

**Figure 6 antioxidants-10-01171-f006:**
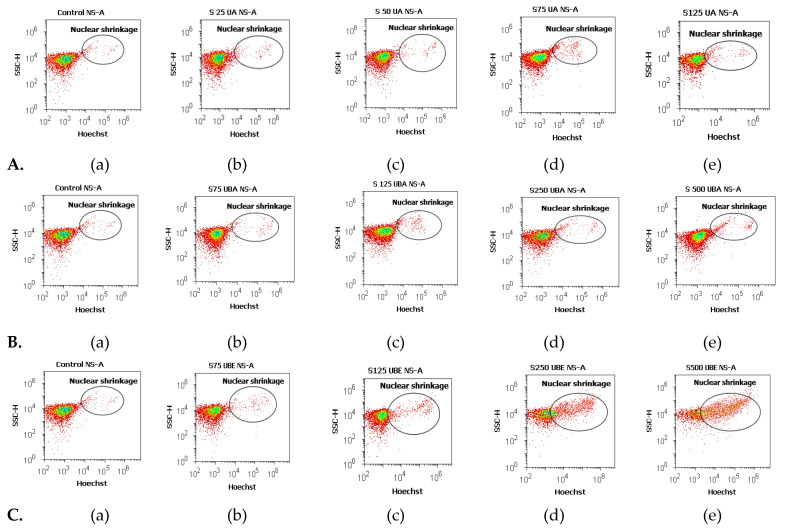
Nuclear shrinkage status of UA, UBA, and UBE treatments in normal blood cell cultures. Hoechst patterns of UA 25, 50, 75, 125 µg/mL **A**(**b**–**e**); **B**(**b**–**e**) UBA 75, 125, 250, 500 µg/mL; **C**(**b**–**e**) UBE 75, 125, 250, 500 µg/mL reported to 1% DMSO (**A**(**a**), **B**(**a**), **C**(**a**)).

**Figure 7 antioxidants-10-01171-f007:**
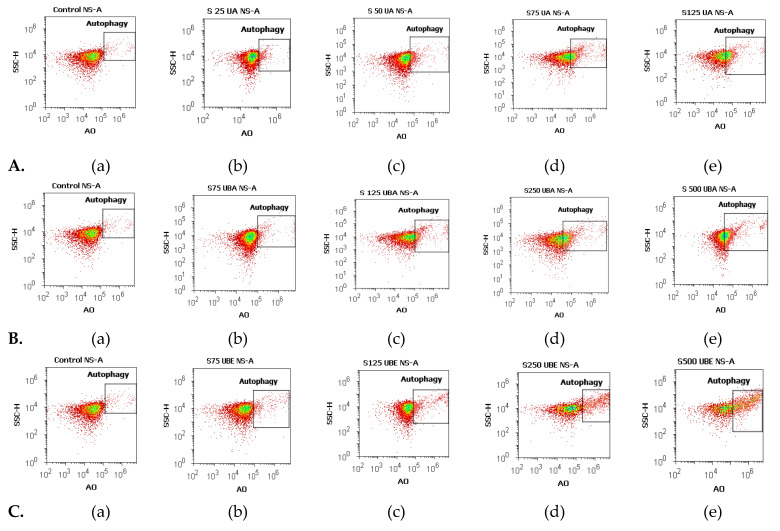
Autophagy status of UBA, UBA, and UBE treatments in normal blood cell cultures. Acridine orange patterns of UA 25, 50, 75, 125 µg/mL **A**(**b**–**e**); **B**(**b**–**e**) UBA 75, 125, 250, 500 µg/mL; **C**(**b**–**e**) UBE 75, 125, 250, 500 µg/mL reported to 1% DMSO (**A**(**a**), **B**(**a**), **C**(**a**)).

**Figure 8 antioxidants-10-01171-f008:**
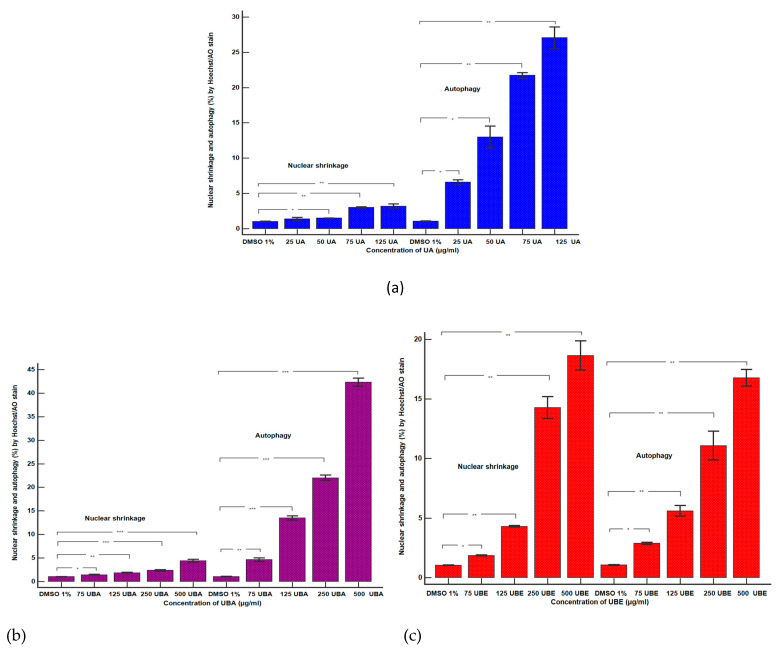
Statistical analysis of nuclear shrinkage and autophagy: (**a**) UA; (**b**) UBA; (**c**) UBE. * *p* < 0.05. ** *p* ≤ 0.01, and *** *p* < 0.001 represent significant statistical differences between the control and samples made by paired samples *t*-test.

**Figure 9 antioxidants-10-01171-f009:**
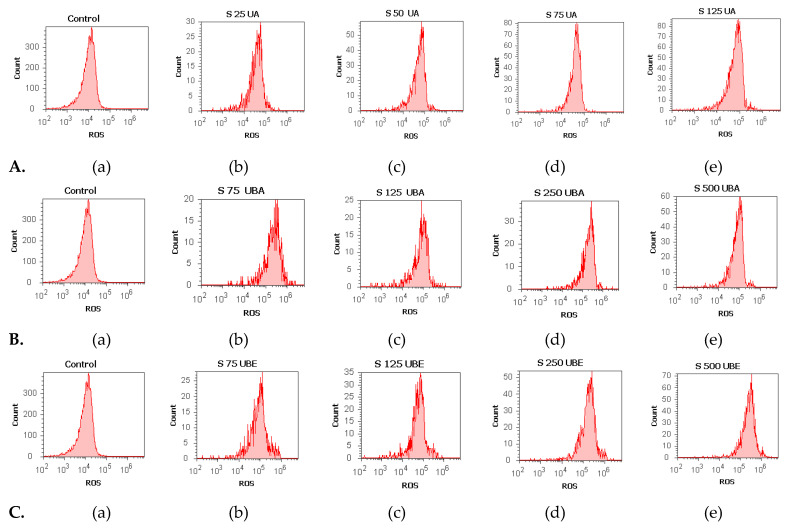
Reactive oxygen species (ROS) status of UA, UBA, and UBE treatments in normal blood cell cultures. ROS patterns of UA 25, 50, 75, 125 µg/mL **A**(**b**–**e**); **B**(**b**–**e**) UBA 75, 125, 250, 500 µg/mL; **C**(**b**–**e**) UBE 75, 125, 250, 500 µg/mL reported to 1% DMSO (**A**(**a**), **B**(**a**), **C**(**a**)).

**Figure 10 antioxidants-10-01171-f010:**
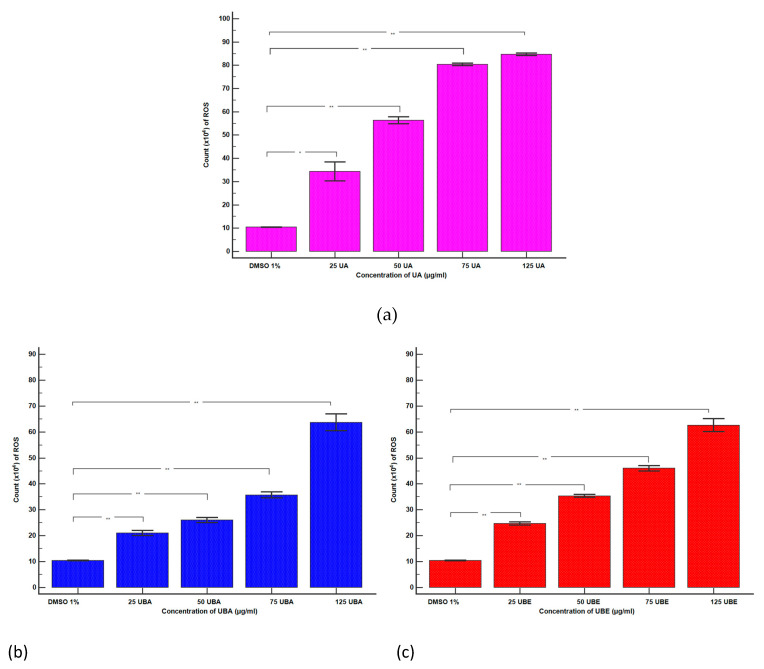
Statistical analysis of cellular oxidative stress: (**a**) UA; (**b**) UBA; (**c**) UBE. * *p* < 0.01 and ** *p* < 0.001 represent significant statistical differences between the control and samples made by paired samples *t*-test.

**Figure 11 antioxidants-10-01171-f011:**
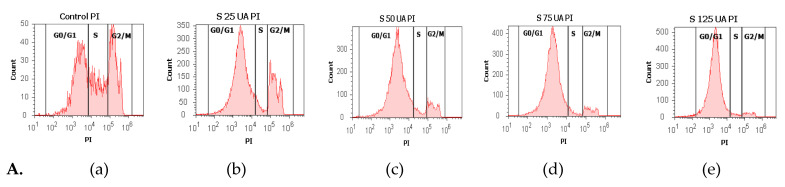
Cell cycle model of UA, UBA, and UBE treatments in normal blood cell cultures. PI/RNase A patterns of UA 25, 50, 75, 125 µg/mL **A**(**b**–**e**); **B**(**b**–**e**) UBA 75, 125, 250, 500 µg/mL; **C**(**b**–**e**) UBE 75, 125, 250, 500 µg/mL reported to 1% DMSO (Negative Control, **A**(**a**), **B**(**a**), **C**(**a**)).

**Figure 12 antioxidants-10-01171-f012:**
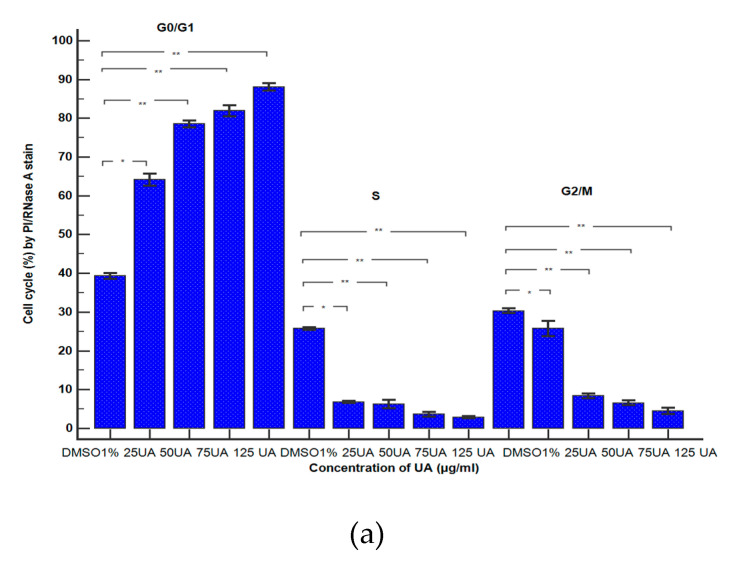
Statistical analysis of DNA content: (**a**) UA; (**b**) UBA; (**c**) UBE. * *p* < 0.01, and ** *p* < 0.001 represent significant statistical differences between control and samples made by paired samples *t*-test.

**Figure 13 antioxidants-10-01171-f013:**
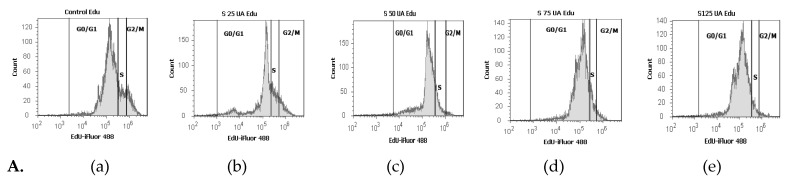
Cell proliferation status of UA, UBA, and UBE treatments in normal blood cell cultures. EdU-iFluor 488 patterns of UA 25, 50, 75, 125 µg/mL (**A**(**b**–**e**)); (**B**(**b**–**e**)) UBA 75, 125, 250, 500 µg/mL; (**C**(**b**–**e**)) UBE 75, 125, 250, 500 µg/mL reported to 1% DMSO (**A**(**a**), **B**(**a**), **C**(**a**)).

**Figure 14 antioxidants-10-01171-f014:**
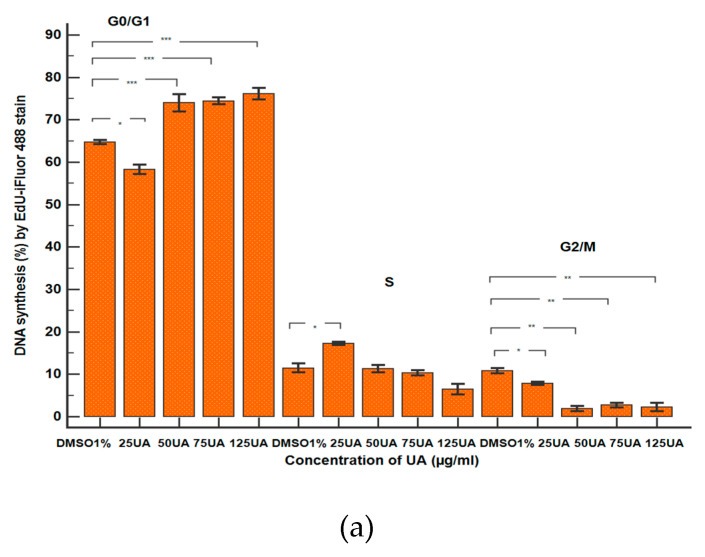
Statistical analysis of DNA synthesis: (**a**) UA; (**b**) UBA; (**c**) UBE. * *p* < 0.05, ** *p* < 0.01, and *** *p* ≤ 0.001 represent significant statistical differences between the control and samples made by paired samples *t*-test.

## Data Availability

Data is contained within the article and Appendix A.

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
