# Peer review of "Usnic Acid and Usnea barbata (L.) F.H. Wigg. Dry Extracts Promote Apoptosis and DNA Damage in Human Blood Cells through Enhancing ROS Levels"

_antioxidants, 2021, doi:10.3390/antiox10081171_

Round 1

Reviewer 1 Report

The paper focusing on the biological properties of purified usnic acid (UA) or the UA-rich extracts as obtained from the lichen Usnea barbata, is an interesting and creative continuation of the authors' work with this natural compound having been used in traditional Chinese medicine for centuries. The unique structure and properties of UA make it possible to be used as a potent anticancer drug, however, thorough investigation of its signaling mechanisms and toxicity relative to both normal and cancer cells, is of critical importance. The authors analyze cell death modes as well as selected signaling mechanisms as observed upon human blood cells treatment with UA. The methods used in the study are solid enough, and the results seem sound and convincing with potential clinical applicability in the future. There are, however, some minor improvements which should be introduced before publication.

The introduction contains a little overly detailed description of relatively well known and obvious cell death mechanisms (lines 86-112) which would better suit for a kind of general cell death review rather than an original paper. As such, I recommend shortening it slightly, and instead discussing the problem of potential cell death as expected in a random blood sample importantly taking into account different types of cells and morphotic elements. It might be of interest to the reader if the authors would discuss the available current literature data pertaining to specific death modes of the red blood cells as well as the white blood cells, preferably also referring to different white blood cell species. Until very recently, erythrocytes have been considered unable to undergo apoptosis, as they lack mitochondria and nuclei, however, according to most recent observations treatment of erythrocytes with the Ca2+-ionophore, ionomycin, leads to cell shrinkage, cell membrane blebbing and annexin binding, all typical features of apoptosis as observed in other cell types. Interestingly, it was found that erythrocytes contain procaspase-3 and procaspase-8 at levels comparable with those found in Jurkat cells, however, in contrast to most other cells, erythrocytes do not contain other elements of the apoptotic machinery, such as Apaf-1, cytochrome c, and caspases-2, -6, -7 and -9.  (Holcik M., Trends in Genetics, 2002, 18(3), P121). Moreover, activation of caspases-8 and -3 was also observed in peroxynitrite-induced apoptosis-like phenomenon in human red blood cells. Application of flow cytometry and fluorescence and confocal microscopy led to demonstrate that apoptosis of the red blood cells is a physiological phenomenon (Bratosin D. et al., Cytometry A., 2009, 75(3):236-44). 

As for Materials and Methods, in section 2.1.2. "Determination of the purity of isolated usnic acid", some variables used in the calculations are not described precisely enough, missing the measurement units (Mstd = standard mass weighed and Ms = sample weighed mass). This part should definitely be reviewed.

Chromatograms from the separation and purity determination steps seem very reliable and convincing.

In Figure 5, 8, 11 and 17, the abbreviation for usnic acid as shown in the graph, should be consequently UA rather than AU. I suggest double checking all of the abbreviations used in presented figures.

I recommend to avoid reporting the interpretation of the statistically insignificant results such as, e.g. the one in lines 513-514: "concentrations of UA (50, 75 and 125 μg/mL) induced decreased values of DNA synthesis without remarkable differences reported to negative control" the way it is presented in the manuscript. It would be more informative to simply phrase it as "concentrations of UA (50, 75 and 125 μg/mL) did not significantly alter DNA synthesis relative to control". I also recommend similar rephrasing for lines 521-523, and the like elsewhere in the paper.

Lines 526-527: I suggest "paired samples t-test" rather than "paired sample T-test" (lowercase t).

Discussion does not address the problem of different types of blood cells present in the full blood sample, just using a very general term "normal blood cells". It would be interesting to comment on the presence of phospatidylserine residues on the surface of erythrocytes after treatment with UA. Also looking at the structure of UA, it seems like it should participate in the Michael addition reactions binding intracellular glutathione which might be the major oxidative stress generation mechanism. Therefore, pretreatment of the cell culture with N-acetylcysteine should significantly reduce the level of oxidative stress in response to UA treatment, and this way prove its causative role in the cell death mechanism signaling. Using some other natural or synthetic antioxidant in order to reverse the prooxidative effect of UA might also be a tempting idea. Importantly, severe depletion of the glutathione level is believed to be the cause of acute liver failure after administration of UA in humans, as reported in the literature listed below:
1. Hsu, LM; Huang, YS; Chang, FY; Lee, SD (2005). "'Fat burner' herb, usnic acid, induced acute hepatitis in a family". Journal of Gastroenterology and Hepatology. 20 (7): 1138–9.
2. Sanchez, William; Maple, John T.; Burgart, Lawrence J.; Kamath, Patrick S. (2006). "Severe Hepatotoxicity Associated with Use of a Dietary Supplement Containing Usnic Acid". Mayo Clinic Proceedings. 81 (4): 541–544.
3. Yellapu RK, Mittal V, Grewal P, Fiel M, Schiano T (2011). "Acute liver failure caused by 'fat burners' and dietary supplements: a case report and literature review". Canadian Journal of Gastroenterology. 25 (3): 157–60.

The discussion might be enriched at least by mentioning the Safety Alerts for Human Medical Products as released by FDA pertaining to Lipokinetix, a compound supplement containing UA. It seems to be of critical importance in context of the study goal emphasized by the authors which very directly refers to the problem of UA safety and toxicity as a potential anticancer drug.

It might also be good to mention UA in the discussion, as a mitochondrial uncoupling agent leading to depletion of ATP synthesis in the cell, and ultimately causing potential necrotic cell death as was observed in some studies with cultured hepatocytes.

According to the available literature, (+) UA gets more attention than (-) UA as it exhibits greater potency and a wider range of biological activities. Therefore, it is kind of logical to answer the question if natural UA as isolated from the lichen exists in the form of a racemic mixture or if there rather is any stereoisomeric preference observed.

In Conclusions section, line 649: I recommend changing "low damage on normal cells" to "relatively lower damage to the normal blood cells". It would be of interest to refer to IC50 concentrations of UA as determined for various cancer cells, and specifically neoplastic white blood cells like the ones isolated from patients presenting with leukemia (e.g. Wang et al., Bioorg Chem., 2019, 89:102971: "Discovery of novel (+)-Usnic acid derivatives as potential anti-leukemia agents with pan-Pim kinases inhibitory activity"). Only then, the concept of high therapeutic index as suggested by the authors would be well supported and become more obvious to the reader.

Reviewer 2 Report

Currently, compounds of natural origin are a very important element of anti-cancer therapy. Hence, tests checking their cytotoxicity to the human body, including blood cells, are very important.

The design and methodology of the research were mostly correctly selected and confirmed the thesis presented. The manuscript is interesting. However, some major issues occur and they need to be addressed

  1. In the introduction, there are long phrases of text from the articles cited, followed by the pages listed. In my opinion, it is better to paraphrase the text so that it is not plagiarism, and not give the pages, because sometimes they do not agree with the pages in the article.
  2. All figures are of poor quality
  3. In figures 2,3,4, the control on plot (a) is the same as 1% DMSO in the graph (f). The names should be standardized so as not to mislead the reader
  4. Please check again the statistical significance between the control and 50 µg/ mL UA, because my doubts are raised by the statistical difference at the level of 1%
  5. In figure 2 b, c, d, e is AU instead of UA
  6. Please correct the description of figure 2 because there is a mistake (‘Annexin V-FITC/PI pattern at 1% DMSO and UA 25, 50, 75, 125 µg/m mL;’)
  7. ‘The obtained data revealed that 75 µg/ mL of UBA determined a slight diminution in cell viability: 80.16±0.57% vs 96.89±0.14%, p<0.01; also, 321 slightly elevated cell apoptosis: 19.45±0.60% vs 2.72±0.16%, p><0.01 (Figure 3, b).><0.01’ and there is no statistical significance here? and the difference is slight. There should be no p <0.001.
  8. I have the same objections to UBE (75 µg/ mL)
  9. Descriptions in figures should agree with the text
  10. Descriptions in the figures should be consistent with the text and all figures should be referenced in the text. Missing e.g. 3c, 3d, etc.
  11. The IC50 is a quantitative measure that indicates the amount of a particular inhibitory substance (e.g. drug) required to inhibit a given biological process or biological component by 50% in vitro. Usually, the MTT test is used for such analyzes, because the inhibition of cell viability must not necessarily be related to apoptosis
  12. Please explain why the level of 3/7 caspase is associated with early apoptosis
  13. What does nuclear apoptosis - late apoptosis? Dyes that bind to DNA, such as Hoechst 33342, can be used to observe nuclear condensation.
  14. I do not see the point in presenting the distribution of the cell cycle phases twice with PI and EdU-iFluor 488, taking into account the fact that the results were different, and they should be similar. How to explain the fact e.g. G0 / G1 phase (1% DMSO) with PI about 40%, and with EdU-iFluor 488 about 65%?
  15. There should also be a peak with apoptotic cells on the plots of the cell cycle
  16. I suggest that you read the article again carefully because there are editorial mistakes, e.g. apopsosis instead of apoptosis

Reviewer 3 Report

Popovici et al. have conducted a study entitled “Usnic acid and Usnea barbata (L.) F.H. Wigg. Dry Extracts promote Apoptosis and DNA Damage in Human Blood Cells through Enhancing ROS Levels”. The authors studied toxic effect of UA, UBA and UBE isolated from UBEA by analyzing apoptosis, caspase activity, autophagy, ROS level and DNA damage on human blood cells. Finally, authors suggested that evaluation of natural products toxicity is also improtant for anticancer drug development. Indeed, data presented by Popovici et al. showed good toxicity of UA, UBA, and UBE. But, there are some issues that need to be addressed.

  1. In the cytotoxic data, I recommed that authors should treated more concentration for determination of LC50. And, please calculate therapeutic index for comparison LC50 and EC50. Then, authors arguments will becom clearer
  2. In Fig. 1, the resolution of chromatogram is very low. I recommend to improve figure quality
  3. Overall, in the results, the authors presented flow cytometric cytogram and analysis. But, I think it is less clear. I recommend to separate them or rearrange the figure.

Round 2

Reviewer 2 Report

All my comments have been implemented. I accept this article in the present form.